# Exploring Spatio-Temporal Variations of Ecological Risk in the Yellow River Ecological Economic Belt Based on an Improved Landscape Index Method

**DOI:** 10.3390/ijerph20031837

**Published:** 2023-01-19

**Authors:** Meirui Li, Baolei Zhang, Xiaobo Zhang, Shumin Zhang, Le Yin

**Affiliations:** 1College of Geography and Environment, Shandong Normal University, Jinan 250014, China; 2Zaozhuang Municipal Bureau of Natural Resources and Planning, Zaozhuang 277099, China; 3Research Institute of Regional Economy, Shandong University of Finance and Economics, Jinan 250014, China

**Keywords:** ecological risk assessment, landscape pattern, spatiotemporal pattern, Yellow River Ecological Economic Belt

## Abstract

Intense human activities have led to profound changes in landscape patterns and ecological processes, generating certain ecological risks that seriously threaten human wellbeing. Ecological risk assessment from a landscape perspective has become an important tool for macroecosystem landscape management. This research improves the framework and indices of the ecological risk assessment from a landscape perspective, evaluates the land use pattern and landscape ecological risk dynamics in the Yellow River Ecological Economic Belt (YREEB), analyzes the spatiotemporal variation, and identifies key areas for ecological risk management. The results indicate the following: The main land use types in the region are grassland and cropland, but the area of cropland and grassland decreased during the study period, and with the accelerated urbanization, urban land is the only land use type that continued to increase over the 20-year period. The ecological risk in the YREEB tended to decrease, the area of low ecological risk zones increased, while the area of high ecological risk zones gradually decreased. Most areas are at medium risk level, but the risk in central Qinghai and Gansu is obviously higher, and there is a dispersed distribution of local high- and low-risk zones. A total of 37.7% of the study area is identified as critical area for future risk management, and the potential for increased risk in these areas is high. These results can provide a basis for sustainable development and planning of the landscape and the construction of ecological civilization in ecologically fragile areas.

## 1. Introduction

With the rapid development of the productivity and material level of human society in recent years, environmental problems have become increasingly serious [1,2]. In 1998, the publication of the “Guide to Ecological Risk Assessment” marked the maturity of ecological risk assessment [3]. Ecological risk assessment (ERA) was then proposed to link human activities to environmental conditions that are likely to have adverse ecological impacts or consequences according to the assessment process [4,5,6,7]. ERA contributes to ecological conservation and management through scientific evaluation of the negative effects caused by a hazard [8,9].

The scope of ecological risk assessment research mainly focuses on the impact of pollutants on the ecosystem and its composition in the early stage [10,11]. In recent years, the assessment of ecological risks and ecosystem values has become a hot research topic [12,13]. There are two main methods used to assess ecological risk. One is a risk source-based hazard assessment model, and the canonical form uses the paradigm of “risk source identification-receptor analysis-exposure and hazard assessment-risk characterization evaluation” [14,15,16,17]. The other method is to establish a landscape pattern based on the occurrence of processes and spatial patterns of landscape ecology, using remote sensing and GIS techniques to analyze ecological risks [18,19,20,21]. The former method of the risk source sink assessment model is a general model for risk assessment, which includes the pressure–state–response (PSR) model [22,23], fuzzy assessment [24] and support vector [25]. However, the promotion and application of such methods are limited in application, due to the lack of field ecological monitoring data, especially in regions of complex spatial heterogeneity [26]. The heterogeneity assessment of ecological risk is an important component of ecological studies due to of its ability to clarify the major ecological factors and processes that govern ecological persistence [9]. It also facilitates qualitative and quantitative research on natural and human factors, and has become one of the hot research areas meant to deal with the integrated management of social–ecological complex systems [27].

The experts proposed an already widely used method for quantifying landscape ecological risk based on landscape disturbance and landscape vulnerability, which is already used in many areas, such as lowlands [28], coastal region [29], ecologically vulnerable areas [30], and large cities [31]. However, these studies vaguely describe and visualize the ecological risks according to specific conditions. Vulnerability analysis, on the other hand, only considers static differences between landscape types [32], which does not fully reflect the spatio-temporal heterogeneity of risk, especially at larger regional or inter-temporal scales [33]. Therefore, the method still needs targeted improvements in terms of practical applications at spatial scales [34], ecological space [35], and management purposes.

The Yellow River Ecological Economic Belt (YREEB) is an important ecological barrier in China [36]. The YREEB is also an important economic region in China, gathering three national urban agglomerations (Central Plains urban agglomerations, Guanzhong urban agglomerations, and Lanxi urban agglomerations) and three regional urban agglomerations (Shandong Peninsula urban agglomerations, Ningxia-Yanhuang urban agglomerations, and Jinzhong urban agglomerations). Due to regional development and unreasonable governance, the ecological barrier function of the YREEB has been seriously degraded [37], and is extremely vulnerable to human and natural factors. In order to integrate it into the overall national strategic development and ecological civilization, the economic development and ecosystem coordination of the YREEB have become the focus of scholars [38]. However, in order to support the environmental management of the Yellow River region, most analyses tend to focus on natural disasters [38], environmental contamination [39], healthcare [40], and other relevant studies. The limited number of studies on the cumulative ecological risks of human activities and environmental change to the surface ecosystems of the Yellow River basin, especially regarding the terrestrial ecosystems, have hindered progress for local environmental management [41].

The main objectives of this study are to: (1) improve the current landscape standard on the basis of an approach to develop a more suitable dynamic model of landscape vulnerability for the study as a way to calculate the Landscape Ecological Risk Index (LERI) in the YREEB from 2000 to 2020 [42]; (2) quantify the spatial and temporal variability and differences in ecological risk in the YREEB landscape; (3) identify ecological key areas of the YREEB for risk management. The study provides a theoretical reference for land use optimization and decision-making for sustainable regional environmental and ecological risk management.

The structure of this paper is as follows. Section 2 presents the study area, data sources, and research methods. Section 3 describes the research results. Section 4 features a discussion. Section 5 draws the main research conclusions and highlights policy implications.

## 2. Materials and Methods

### 2.1. Study Area

The YREEB is located in the central part of China, covering Qinghai, Gansu, Ningxia, Shanxi, and Shaanxi, as well as Henan and Shandong (Figure 1). The total area of the YREEB is 1,863,000 km^2^. The region has complex and diverse landform types. It has a continental climate, and the land use types are mainly grassland, arable land, and forest land. In 2020, grassland occupied the largest area, accounting for 40.3% of the total area of the Yellow River Ecological Economic Belt, mainly concentrated in Qinghai and Gansu [43]. The Qilian Mountains are distributed in the junction zone of two provinces, which is a zone of mountain landforms, Danxia landforms, river landforms, and glacial tundra landforms. At the same time, the region has the largest inland saltwater lake, called the China–Qinghai Lake, which is a closed lake mainly recharged by precipitation. The region has sufficient light, low rainfall, and distinct dry and wet seasons. In 2020, the total GDP and resident population of the region was about 1.87899 billion yuan and 106.49 million people, respectively [44,45].

### 2.2. Data Sources

The land use data, which is the basic data for this study, is Landsat TM/OIL visual interpretation data with a resolution of 30 m. Using the function of randomly selected points of ArcGIS to check, the accuracy of the decoding results were obtained at 94.5% [46]. The landscape types are classified into six categories: cropland, woodland, grassland, water, urban land, and unused land. Other data required to calculate ecological risk are shown in Table 1.

### 2.3. Methods

In order to facilitate the follow-up study, the study area is divided into 20 km according to the actual scope of the YREEB, referring to existing studies, 5048 ecological risk zones are obtained from a 20 km square grid using ArcGIS fishnet analysis and statistical tools, and then the ecological risk index is calculated for each grid. Secondly, the improved landscape ecological method is used to calculate the ecological risk index of each grid unit in 2000, 2005, 2010, 2015, and 2020. Finally, the model and dynamics of ecological risk from 2000 to 2020 are analyzed, and the key areas of risk management are discussed. The specific steps are shown in Figure 2.

#### 2.3.1. Landscape Ecological Risk Index Model Construction

The construction of a landscape ecological risk index is the key to quantitative analysis of landscape ecological risk [47]. In this study, the landscape pattern change approach is chosen to calculate the landscape ecological risk index, and landscape pattern change is one of the most direct manifestations of human activities on land use [48]. Since different landscapes differ in their resistance to external influences, the ecological landscape risk model of the study area was constructed from the perspective of landscape structure, components, disturbance index, and vulnerability index [49,50]. The calculations are as follows:(1)$$LERI_{k}=\sum_{i=1}^{n}\sqrt{E_{ki}\times F_{ki}}\times \frac{A_{ki}}{A_{k}}\;$$
where $LERI_{k}$ is the landscape ecological risk index of the *k*th risk plot, $E_{ki}\;$ is the landscape disturbance index of landscape type *i* in the *k*th risk plot, $F_{ki}$ is the landscape vulnerability index of landscape type *i* in the *k*th risk plot, $A_{ki}$ is the area of landscape type *i* in the *k*th risk plot, and $A_{k}$ is the area of the *k*th risk plot. The landscape disturbance index ($D_{ki}$) is calculated by landscape fragmentation ($C_{ki}$) [49], landscape separation ($S_{ki}$) [51] and landscape dominance ($D_{ki}$) [52]. The vulnerability index ($F_{ki}$) is calculated using as an empirical value ($EV_{i}$) and a composite adjustment factor ($MN_{k}$) [53]. The used landscape indexes and their implications are shown in Table 2.

#### 2.3.2. Analysis of the Spatial and Temporal Variation of LERI

The spatio-temporal variation of LERI is analyzed in three main aspects. First, the ecological risk level is classified into five levels using the Natural Breaks method [63]. The second is the construction of the rate of change of risk index [64], which includes the identification of spatial and temporal heterogeneity in risk variation by comparing the differences in growth rates between units over time. Finally, the stability of ecological risk was evaluated by calculating the coefficient of variation of LERI [65].

The risk rate of change index is the annual increase per unit of LERI as a percentage of the initial LERI value. A higher value indicates a faster increase in risk and a negative value indicates a decrease in risk. The calculation formula is:(2)$$RRC_{k}=\frac{(LERI_{k}^{T2}-LERI_{k}^{T1})}{LERI_{k}^{T1}}\times \frac{1}{\Delta T}\times 100\%\;$$ 
where $RRC_{k}$ is the rate of risk change of the *k*th risk plot; $LERI_{k}^{T2}$ and $LERI_{k}^{T1}$ are the landscape ecological risk index of the *k*th risk plot at time *T*1 and *T*2, respectively; and Δ*T* is the time span from *T*1 to *T*2.

In general, the larger the coefficient of variation, the weaker the risk stability [66]. The coefficient of variation is calculated as follows:(3)$$CFV_{k}=SD_{k}/AV_{k}\;$$
where $CFV_{k}$ is the coefficient of variation in the LERI in the *k*th risk plot, $SD_{k}$ is the standard deviation of the LERI in the *k*th risk plot, and $\;AV_{k}$ is the mean of the LERI in the *k*th risk plot.

## 3. Results

### 3.1. Evolution of Land Use Landscape Pattern

There were considerable differences in the area and change trajectories of each land use type across the study period. The land use change in the region from 2000 to 2020 is shown in Figure 3a and the land use transfer matrix from 2000 to 2020 is shown in Figure 3b. The results show that the main land use types are grassland and cropland. Water, woodland, and urban land showed an increasing trend, while cropland and grassland showed a decreasing trend. During 2000–2020, the area of grassland decreased by 4500.7 km^2^, and the area of cropland decreased by 39,756.7 km^2^, respectively. Urban land was the only type of land use that had continued to increase during 2000–2020. Compared with 2000, the area of urban land increased by 1.55 times, with a net increase of 26,167 km^2^. The increased urban land mainly came from cropland, woodland, and water, and the converted areas of the above three were 24,497.5 km^2^, 1334.11 km^2^, and 595.9 km^2^, respectively. On the whole, the YREEB’s land use pattern has changed significantly over the past 20 years, and the conversion from cropland to urban land was the main land use change mode in the YREEB. With the development of urbanization, the phenomenon of urban land occupying arable land is particularly serious, and if the supplementation of arable land by other land types is not considered, the rate of arable land reduction will continue to increase. While developing cities, attention should be paid to the use and protection of cropland.

### 3.2. Spatiotemporal Dynamics of Landscape Ecological Risk

#### 3.2.1. Spatial Patterns of Landscape Ecological Risk

The improved landscape ecological risk model was used to calculate the LERI of 5048 evaluation units in five periods and to find the LERI of the YREEB and each province respectively (Figure 4). The average LERI values of the YREEB in 2000, 2005, 2010, 2015, and 2020 were 1.3443, 1.3275, 1.3142, 1.3107, and 1.2180, respectively. This leads to no particularly significant change in the value at risk for the region, but there was a significant downward trend. The provincial-scale value showed a similar trend. The overall ecological situation in most provinces improved significantly from 2000 to 2015, as evidenced by decreasing risk, and then stabilized after 2015. The provinces of Shanxi, Ningxia, and Qinghai had relatively higher risks (Figure 4). The overall landscape structure of these regions were relatively fragmented, and the stability and resilience of ecosystems were poor. The risk to Shandong and Henan were relatively low, which indicated that the landscape resilience of these regions was better than that of the western regions. The ecological risks of Shandong and Henan showed an increasing trend from 2000 to 2010, indicating that the ecological protection and development were ignored while developing the economy in this period, which increased the vulnerability of the environment and the ecological risks. The overall risk of most provinces in this period was reduced, indicating that the ecological situation was significantly improved, thus reducing the ecological risks. Since 2010, each province has basically shown a stable development trend.The ecological risk index is divided into five grades by using the Natural Breaks method: low ecological risk (LERI < 0.60), medium-low ecological risk (0.60 ≤ LERI < 1.18), medium ecological risk (1.18 ≤ LERI < 1.72), medium-high ecological risk (1.72 ≤ LERI < 2.40), and high ecological risk (LERI > 2.4). The proportion of the risk index of each region is obtained and shown in Figure 5. The proportion of high ecological risk areas for the entire region has decreased from 2000 to 2020, from 9.5% to 7.4% (Figure 5a), and the number of units has decreased by 103. The proportion of low ecological risks remained almost unchanged, while the proportion of medium and low risks continued to increase. The trend of risk structure for each province was similar to that for the whole YREEB. The proportion of high risk categories in Qinghai, Gansu, and Shaanxi generally declined, while the proportion of low- and medium-risk areas increased (Figure 5b–h). The proportion of high-risk categories in Ningxia increased first and then decreased. The proportion of medium- and high-risk categories was relatively high (Figure 5d). The proportion of high-risk categories in Shanxi decreased first and then increased, reaching the highest amount in 2010, but only 1.1%. However, the proportion of medium- and high-risk categories in Shanxi was the largest in all regions, reaching 65.5% in 2000. The proportion of low- and medium-low-risk categories was the lowest in all regions (Figure 5f). The proportion of medium- and low-risk categories in Henan was the highest in all regions, with a value higher than 58% (Figure 5g). It can be concluded from the average number of units with different risk levels in each province that the regions with a high proportion of high risks were mainly concentrated in Gansu and Qinghai, and low-risk and medium-low-risk units are mainly concentrated in Henan and Shandong (Figure 5i). Henan Province had no high-risk areas, and Ningxia was dominated by medium- and high-risk units. However, due to the small size of the province, the impact on changes in the overall YREEB risk structure was weak.At the grid scale, the low-risk areas are generally widely distributed, mainly in Yellow River estuary areas of Shandong and Henan province (Figure 6). Human activities in the above areas were relatively infrequent, and the loss degree after human interference and the landscape interference index was small. In addition, the land use type in this area was mainly cultivated land with high land cover continuity, so the vulnerability and fragmentation of this landscape was low. The risks in the north and south sides of Qinghai were also low, because the main dominant landscapes in this area were large areas of grassland and unused land, with less human interference, good vegetation growth conditions, and good ecological protection. The medium-risk areas were mainly distributed in the middle of the study area in blocks, and only scattered in the southwest and northeast, which are mostly located in the transition area between the lower-risk areas and the higher-risk areas. The medium- and high-risk areas are relatively concentrated, mainly located in the western region, including the areas around Qinghai Lake, Qilian Mountain, and southern Gansu, where the surface vegetation was sparse, the biodiversity is low, and the land use type is single. The spatial change of ecological risk had a typical zone transition, which showed that the relatively medium-risk area appeared in the periphery of the relatively low-risk area and the relatively high-risk area, forming a gradient change of ecological risk. From 2000 to 2020, low-risk and medium-low-risk areas expanded, especially in Shanxi and Shaanxi.

#### 3.2.2. Spatiotemporal Differences in Risk Changes

The difference of growth rate between units in different periods is shown in Figure 7, and the average RRC of the entire YREEB was −0.06%. During 2000–2020, the ecological risk at the provincial level showed a downward trend, except in Shandong province, and the ecological risk in Shaanxi, Shanxi, and Ningxia declined faster than that in other provinces. The average RRC value of Henan, Gansu, and Qinghai was higher than that of the YREEB, indicating that the risk reduction rate of these four provinces was slower than that of the entire study area. During 2000–2005, the ecological risk at the provincial level showed a declining trend, except for Shandong province, and the average change rate of the RRC value of Shandong reached 0.10%. The risk in Ningxia declined the fastest, with the average change rate of the RRC value reaching −0.15%. During 2005–2010, the average RRC value changing rate of Shandong, Henan, and Qinghai increased by 0.07, 0.02. and 0.003%, respectively, while the ecological risk in other regions decreased. During 2010–2015, the average change rate of the RRC value of Shanxi, Gansu, and Ningxia was positive, indicating that the ecological risk value of these three regions was on the rise. During 2010–2015, the ecological risk of the YREEB declined the slowest, with an average change rate of the RRC value of −0.003%. During 2015–2020, the risk of the YREEB declined the fastest, with the change rate of the RRC value being −0.003%. During 2015–2020, the Shanxi, Shaanxi and Shandong had a higher average change rate of the RRC value, and the change rates were all below -0.15%. In addition, the regions with the most significant changes in each period are Shandong, Ningxia, and Shanxi in the first stage; Shaanxi and Shanxi in the second stage; Shandong, Gansu, and Ningxia in the third stage; and Shanxi and Shandong in the fourth stage.

Changes in ecological risk were unevenly distributed. From 2000–2020, most regions reduce risk at a rate of 0–0.05% per year (Figure 8). The rest of the units with slightly increased risks were mainly concentrated in the south and north of Qinghai, and the north of Gansu (Figure 8a). During 2000–2005, the risk reduction units were mainly distributed in the central region, mainly in most regions of Shanxi, Ningxia, and Shaanxi, especially in the northwest of Gansu (Figure 8b). During 2005–2010, the units of increased risk gradually decreased, with uneven and sporadic distribution, mainly scattered in the northern part of Qinghai (Figure 8c). During 2010–2015, the regions where the risk decreased obviously showed a trend of gradual diffusion from the middle to the four sides, and the regions with rapid decline appeared in western Shandong, central Henan, and southern Qinghai (Figure 8d). During 2015–2020, the number of risk growth units increased significantly, and were concentrated in southern Qinghai, southern Shaanxi, and small adjacent areas with increasing risks in Gansu (Figure 8e). In general, the change rate of risk growth units was generally lower than that of risk decline units, so the overall risk in the study area had not increased significantly.

### 3.3. Key Areas for Ecological Risk Management

Based on the actual situation of the YREEB, the dynamic characteristics of ecological risks, and the stability of ecological risks, key areas for future risk management were identified. In this regard, ecological risk management is divided into three types of key areas: stable high-risk areas, risk reduction-lagged areas, and risk-unstable areas. Stable high-risk areas are areas where risk levels remained high for the duration of the study period; therefore, these areas pose a serious threat to ecological safety and environmental health, which is usually a concern for risk managers. Risk reduction-lagged areas are areas where the rate of risk reduction is slow, indicating that the environmental protection measures implemented by managers may be flawed. Risk-unstable areas are areas where the level of risk is unstable under the influence of changing environmental conditions, and therefore have a greater likelihood of future risk increases.

In order to accurately locate the critical areas, multiple risk management scenarios are designed based on expert opinions under the premise of satisfying and optimizing the control objectives, and finally, the thresholds of LERI, RRC, and CVR are determined as 15%. The stable high-risk areas numbered 397 units, which were relatively concentrated. Most of them are distributed in the west of the YREEB, mainly in Qinghai and the border area between Qinghai and Gansu, and form two clusters: the Qinghai Tibet Plateau and the Qaidam Basin (Figure 9a). Risk reduction-lagging areas numbered 720 units, 65% of which overlap stable high-risk areas (Figure 9b), and the rest of the units were mainly located in a small part of Ningxia and Shandong provinces, and were scattered in Shaanxi, Shanxi, and other places as well. The risk unstable areas numbered 786 units, had almost no overlap with the first two types, and are scattered in the provinces and regions (Figure 9c). When these three types were combined, there were 1903 risk management key units, accounting for 37.7% of the study area (Figure 9d). Qinghai Province is the most critical region, followed by Gansu Province and Shaanxi Province.

## 4. Discussion

### 4.1. Spatiotemporal Differentiation of Factors Influencing LERI

In this study, the landscape ecological risk index evaluation model was improved on the basis of the original one to analyze the temporal and spatial dynamics of land use and the differences of landscape ecological risk in the YREEB. The conversion from cultivated land to construction land was the main land use change mode in the YREEB, which has been confirmed by relevant studies [67]. At present, the YREEB is still in a period of rapid urbanization [68]. The overall landscape pattern of the YREEB has been broken, ecological risks are intensified, and the pressure on the ecological environment has been increased due to extensive development, unreasonable industrial structure, and disorderly expansion of construction land [69]. There has been an obvious increase in the total area of low- and medium-risk areas and an obvious reduction of the total area of high-risk areas, and the overall landscape ecological risk in the YREEB generally declined. This research is inseparable from the rational use and planning of ecological resources by the Chinese government and similar studies [70]. In order to curb the ecological deterioration of the YREEB, China has successively launched ecological protection and restoration projects such as small watershed management, water and soil conservation, natural forest protection, returning grazing land to grassland, returning farmland to the forest (grassland), slope farmland improvement, wetland protection, etc. [71]. Finally, the vegetation coverage of the YREEB has trended upwards overall over the past 20 years [72].

In the YREEB, regional development imbalance is prominent, and the differentiation of economic development, level of urbanization and policy interventions differentiate the evaluation of ecological levels in the YREEB. The economic development and urbanization level of downstream cities such as Shandong and Henan are higher than most of the cities in the middle and upper reaches [73]; however, the ecological risk level of more developed provinces is mostly low-risk level, and this result is somewhat inconsistent with previous studies [74]. The reason is that the study area in the previous study is only a small part of Shandong along the Yellow River, and that study has considered the impacts of surface morphology and socio-economic factors. Our findings also showed that the medium- and high-risk areas are relatively concentrated in the western region around Qinghai Lake, Qilian Mountain, and southern Gansu, with sparse surface vegetation, low biodiversity, and single land use type, which is basically consistent with the research results of a previous study [75].

### 4.2. Implications for Management

Ecological risk assessment is inseparable from government policy interventions [9]. Recent studies on economic innovation in the Yellow River region have shown that that the industrial structure of provinces and regions along the Yellow River is generally biased, and new momentum for development is insufficient. For upstream regions, it is necessary to systematically protect and repair the ecological environment of key regions and rivers and lakes [76,77]. As for low-risk areas in Qinghai, Gansu, and other places, it is necessary to strengthen the organic linkage between various landscape types, improve biodiversity, and constantly improve the anti-interference ability [78] for its relatively uniform landscape types and fragile ecological baseline. For low- and medium-risk areas, measures should be actively implemented to improve vegetation coverage, reduce landscape fragmentation, and improve landscape integrity and continuity [79]. For higher- and high-risk areas, measures should be encouraged to place windbreakers, return farmland to forests and grasslands, strengthen vegetation coverage on bare land, and prevent water and soil loss and land desertification [80]. In addition, measures should be taken to establish multiple ecological risk buffer zones, strengthen the supervision of the ecological environment, and improve the feedback efficiency of ecological survey results.

The YREEB is still in a period of rapid urbanization [81]. The rapid development of urbanization leads to the intensification of landscape fragmentation, which makes landscapes vulnerable to external invasion and interference, and increases the potential for ecological risk. Therefore, the local government should pay attention to the rational use of land, the fragmentation of landscape patches, and the protection of existing forest land and grassland vegetation, especially in the Qaidam Basin region, during the process of rapid urbanization. For the key areas of risk management in the future, the local government should strengthen the leadership and supervision role of the government for risk management, establish a risk-management system according to key regions, give priority to high-risk areas and other key areas through appropriate policy guidance and financial support, and further improve ecological compensation [82]. Key protected areas across administrative regions should establish regional coordination mechanisms to solve problems together through cooperation [83]. In addition, risk management should be included in future development planning, such as land space planning and ecological restoration planning.

### 4.3. Limitations and Future Research

Through the study of the distribution pattern of landscape ecological risks in the region over the past 20 years, it is found that there has been a significant spatial and temporal divergence in the region, which is the result of the combined effect of the natural environment and human and economic factors in the YREEB. Although the effects of GDP, NPP, NDVI, elevation, slope, population density, temperature, and precipitation were taken into account in the analysis of ecological risk vulnerability, this study is mainly dependent on the current status of land use. When selecting indicators, other types of indicators that are more practical and tailored to local issues can be subsequently added. Therefore, an integrated analysis from multiple natural, social, human, and economic perspectives is necessary in the ensuing assessment and prediction of LERI [84]. On the time scale, it is not possible to make multi-period trend changes, but only spatially and between each geographical element, and the follow-up can make multi-period change studies, while studying the driving relationships between the factors. Considering the early warning and prevention of risks, the subsequent evaluation can be carried out according to the actual situation of each province to give a more definite evaluation of the resilience and prevention ability of the risks in each place. Although the calculation methods and ecological significance of the parameters in the LERI model have been clarified through extensive research on the LERI model in recent years [85], the methodological system of ecological risk management is not yet sound, and the research work is still at the stage of qualitative description, mostly based on the evaluation results of ecological risks and corresponding risk management recommendations [86,87]. Therefore, establishing and improving the basic framework of ecological risk management is an important research direction. Due to the great potential benefits of risk management, work on improving the use of ecosystem services and ecological risks for cross-cutting studies will be important for future research [28]. In the future, efforts could be made in these directions to make the calculation of ecological risks more scientific and rational.

Dividing ecological risk assessment units is the basis for diagnosing regional ecological risks. The grid-based division method is conducive to the expression of spatial heterogeneity of ecological risk, and is a widely used division method at this stage. Most scholars study ecological risks by dividing grid units when studying the scale of medium and small watersheds or the scale of cities and counties [88]. The division method based on administrative regions can assist decision makers in formulating more appropriate risk prevention and control and management policies [89]. It can be seen from the above analysis results that the spatial distribution and change trend of ecological risks of the two scales are similar, each of which confirm the other, thereby improving the scientificity of the ecological risk assessment results. The grid scale is more microscopic and more sensitive to changes in ecological risk, and the division of cells has an important influence on the final risk evaluation results due to the existence of scale effects [90]. Although this study analyzed the landscape ecological risk of the Yellow River Ecological Economic Belt for 20 years, the prediction of future landscape ecological risk needs to be carried out in depth.

## 5. Conclusions

Due to the rapid land use changes in recent years, the impacts on landscape patterns and ecological processes are high, threatening the sustainable development of the region. In this paper, we propose a model different from previous models for calculating ecological risk, establish a new landscape ecological risk model with improved vulnerability, use multiple indicators to assess ecological changes in the region, and use the model to identify key areas for ecological risk management, demonstrating a new perspective that is different from the previous simple construction of ecological risk indices. Additionally, we assessed the ecological value of the YREEB from 2000 to 2020 in spatial and temporal terms, as well as assessed the heterogeneity and stability of landscape ecological risk. The results show that the Qinghai region in the western part of the study area was the highest risk area from 2000 to 2020 due to its fragile ecological environment. Meanwhile, clusters of medium and high risk were also found in Shanxi and Shaanxi, indicating that urbanized human activities had led to increased fragmentation of various landscapes, resulting in increased ecological risk in the region. Over time, the ecological risk in the YREEB showed an overall decreasing trend, with a 2.1 and 3.1% decrease in the high-risk and medium-high-risk areas, respectively, indicating strong measures in the region. Currently, 37.7% of the study area comprises a more urgent risk management status, the local landscape ecological risk is only temporarily mitigated, and the situation of increasing ecological risk has not fundamentally changed. In order to maintain the stability and ecological security of the YREEB, it is necessary to actively optimize land use and take corresponding engineering and biological measures to reduce regional ecological risks and ensure regional ecological security.

## Figures and Tables

**Figure 1 ijerph-20-01837-f001:**
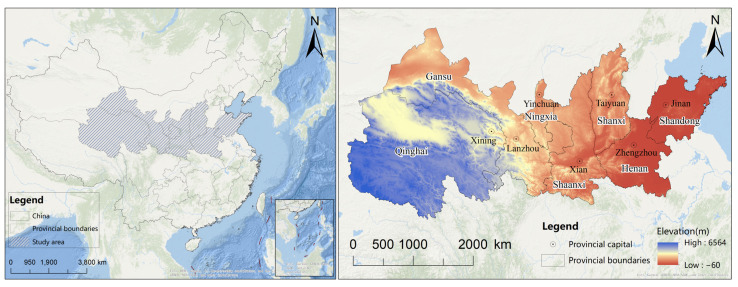
Location of the Yellow River Ecological Economic Belt Area.

**Figure 2 ijerph-20-01837-f002:**
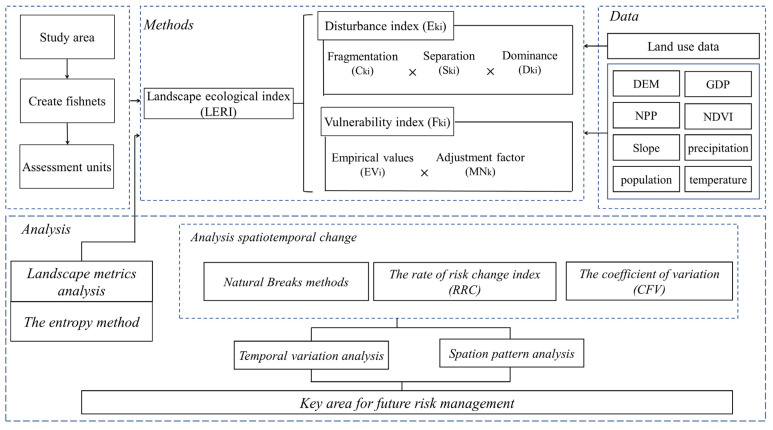
An analytical framework based on an improved ecological risk in the landscape.

**Figure 3 ijerph-20-01837-f003:**
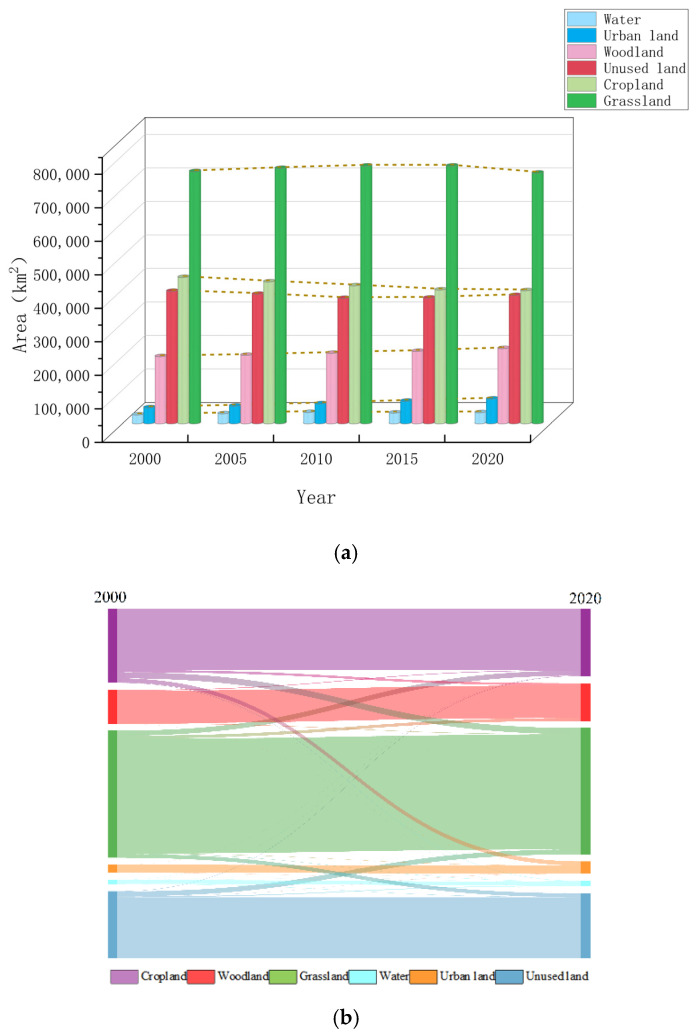
(**a**) Area of each landscape type in the YREEB from 2000 to 2020, (**b**) Transfer matrix of various landscape types from 2000 to 2020 in the YEREEB (unit: km^2^).

**Figure 4 ijerph-20-01837-f004:**
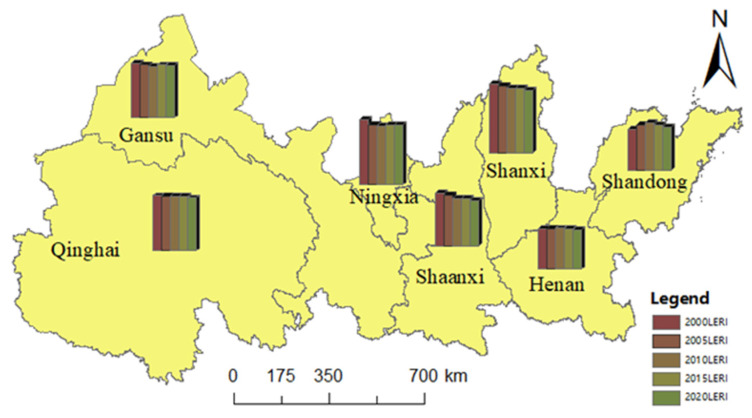
Average LERI values for provinces from 2000 to 2020.

**Figure 5 ijerph-20-01837-f005:**
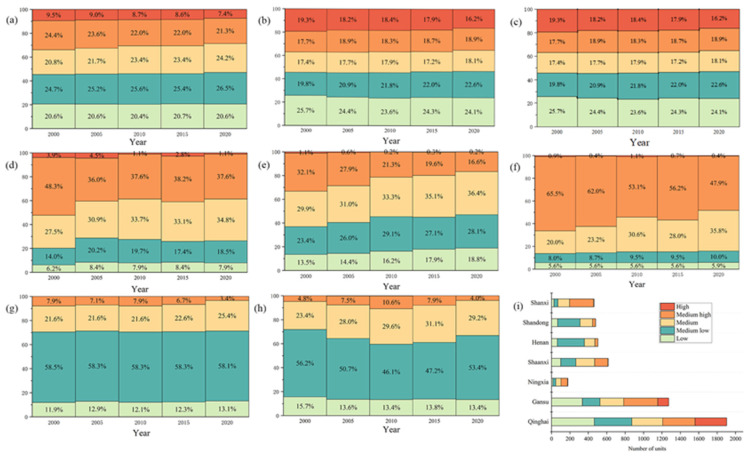
(**a**–**h**) Risk structure for the five time periods. (**i**) The average number of units at different risk levels. Note: YREEB (**a**), Qinghai (**b**), Gansu (**c**), Ningxia (**d**), Shaanxi (**e**), Shanxi (**f**), Henan (**g**), Shandong (**h**).

**Figure 6 ijerph-20-01837-f006:**
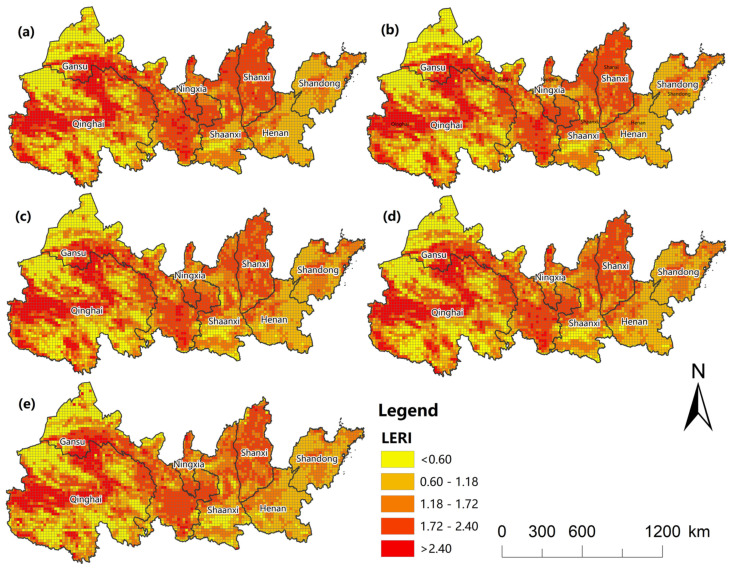
Maps of landscape ecological risk levels in 2000 (**a**), 2005 (**b**), 2010 (**c**), 2015 (**d**), and 2020 (**e**).

**Figure 7 ijerph-20-01837-f007:**
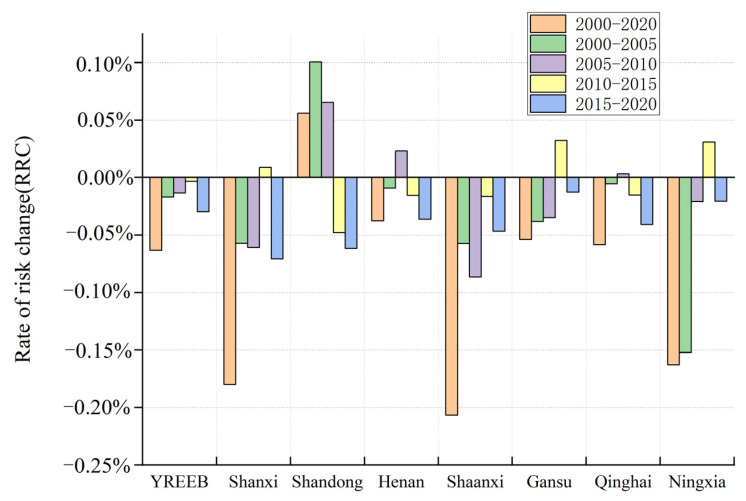
The rate of risk change for the YREEB and provinces in different periods.

**Figure 8 ijerph-20-01837-f008:**
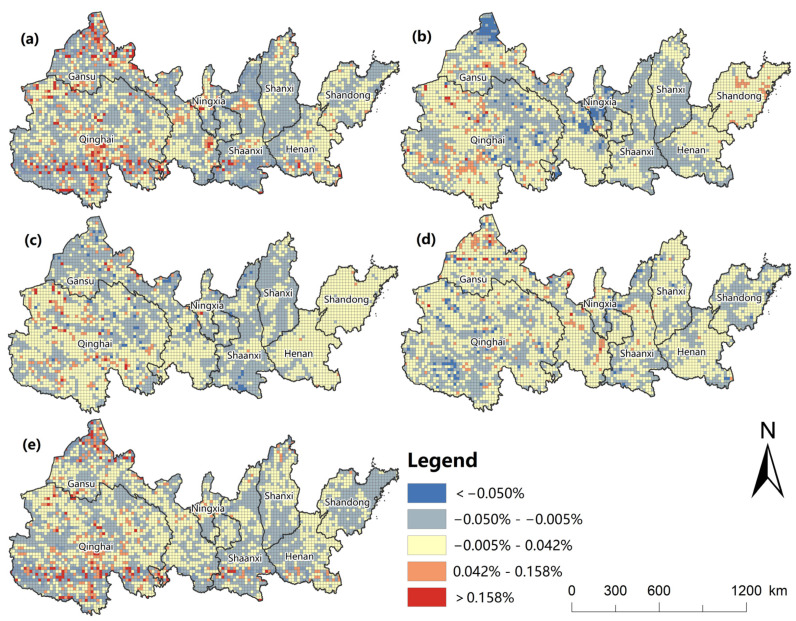
Risk rate of change spatial shows during 2000–2020 (**a**), 2000–2005 (**b**), 2005–2010 (**c**), 2010–2015 (**d**), and 2015–2020 (**e**).

**Figure 9 ijerph-20-01837-f009:**
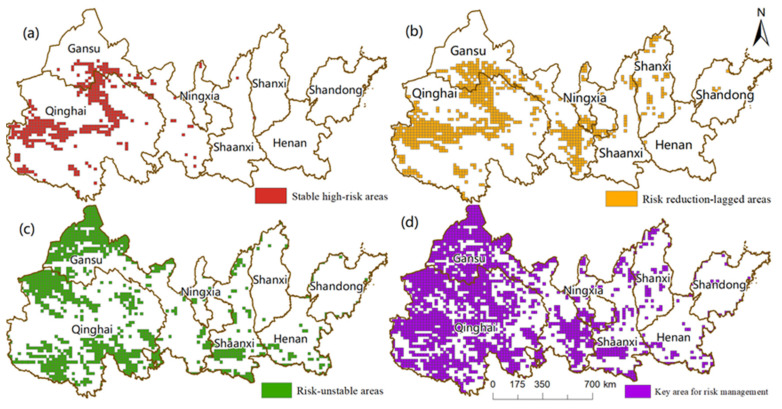
The spatial configuration of risk management. stable high-risk areas (**a**), risk reduction-lagged areas (**b**), risk-unstable areas (**c**), and key areas for risk management (**d**).

**Table 1 ijerph-20-01837-t001:** Data used for the study and sources.

Data	Resolution	Weight	Unit	Sources
Land use	30 m	/	/	https://www.resdc.cn/Default.aspx (accessed on 1 June 2022)
GDP	1000 m	0.2421	10^4^ yuan/km^2^	http://www.geodata.cn (accessed on 2 June 2022)
Population	1000 m	0.1467	Pop/km^2^	https://www.resdc.cn/Default.aspx (accessed on 2 June 2022)
Elevation	30 m	0.1482	m	http://www.resdc.cn (accessed on 4 June 2022)
Slope	30 m	0.0985	%	http://www.resdc.cn (accessed on 4 June 2022)
Temperature	1000 m	0.0892	°C	http://data.cma.cn (accessed on 5 June 2022)
Precipitation	1000 m	0.0647	Mm	http://data.cma.cn (accessed on 6 June 2022)
NPP	1000 m	0.1023	g·C/m^2^	http://modis.gsfc.nasa.gov (accessed on 7 June 2022)
NDVI	1000 m	0.1083	/	https://www.usgs.gov (accessed on 7 June 2022)

**Table 2 ijerph-20-01837-t002:** Landscape indexes and their implications.

Landscape Index	Equation	Implications
Landscape disturbance index ($E_{ki}$)	$E_{ki}=aC_{ki}+bS_{ki}+cD_{ki}$	$E_{ki}\;$ represents a quantitative expression of the magnitude of disturbance to different landscapes within the study area [54], variables *a*, *b*, and *c* represent the weights of $C_{ki}$ $S_{ki}$ and $D_{ki};$ *a* = 0.5, *b* = 0.3, *c* = 0.2
Landscape fragmentation index ($C_{ki}$)	$C_{ki}=\frac{n_{ki}}{A_{ki}}$	$C_{ki}\;$ describes the fragmentation of a continuous large area of land use type into smaller patches after being disturbed by human or natural factors [55]. $n_{ki}$ is the number of patches of landscape type i in the *k*th risk plot, and $A_{ki}$ has the same definition as those given above.
Landscape separation index ($S_{ki}$)	$S_{ki}=\frac{A_{k}}{2A_{ki}}\sqrt{\frac{n_{ki}}{A_{k}}}$	$S_{ki}$ reflects the degree of separation or isolation between land use patches [50], the $n_{ki}$, $A_{ki}$ and $A_{k}\;$ have the same definitions as those given above.
Landscape dominance index ($D_{ki}$)	$D_{ki}=\frac{Q_{i}+M_{ki}}{4}+\frac{L_{ki}}{2}$	$D_{ki}\;$ indicates the dominant landscape of land use types [56,57], where $\;Q_{i\;}$ is the total number of samples in which patch *i* occurs, $M_{ki}\;$ is the number of patch *i* to the total number of patches in the *k*th risk plot, and $L_{ki\;}$ is the total area of patch *i* to the total area of the *k*th risk plot.
Landscape vulnerability index ($F_{ki}$)	$F_{ki}=EV_{i}\times MN_{k}$	$F_{ki}\;$ captures a quantitative representation of the degree of stability of a land use type, indicating the resilience of a landscape type when it is affected by external factors or disturbances by external forces [57]. $EV_{i}\;$ is the empirical value of landscape vulnerability of land use type i. Quantification of indicators by assigning values to different land use types through the expert scoring method: unused land = 6, water = 5, cropland = 4, grassland = 3, woodland = 2, and urban land = 1 [58,59]. $MN_{k}$ is an adjustment factor of the *k*th risk plot.
Compound adjustment factor ($MN_{k}$)	$MN_{k}=EQ_{k}\sqrt{EQ_{k}}$	$MN_{k}\;$ is the compound adjustment factor of the *k*th risk plots. $MN_{k}$ is a composite adjustment factor reflecting the spatial and temporal heterogeneity of landscape vulnerability.$\;EQ_{k}\;$ is the weighted sum of indicators in the *k*th risk plot.
Weighted sum of indicators ($EQ_{k}$)	$EQ_{k}=\sum_{j=1}^{8}w_{j}\times m_{jk}$	$EQ_{k}\;$ is the weighted sum of indicators in unit *k*, $w_{j}$ is the weight of indicator *j*, and $m_{jk}$ is the standardized index value. The study mainly selected eight indicators (*j* = 8), GDP, population density, elevation, slope, temperature, precipitation, NPP, and NDVI [60,61], to comprehensively characterize the vulnerability of the ecological environment, and the weights are determined using the entropy value method [62].

## Data Availability

Not applicable.

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
