# Peer review of "Exploring Spatio-Temporal Variations of Ecological Risk in the Yellow River Ecological Economic Belt Based on an Improved Landscape Index Method"

_ijerph, 2023, doi:10.3390/ijerph20031837_

Round 1
Reviewer 1 Report
In this paper, an improved landscape ecological risk model is applied to analyze the spatial and temporal changes of landscape ecological risk in the Yellow River Ecological Economic Belt from 2000 to 2020. In general, the methods used to calculate the landscape ecological risk dynamics and risk change rate indices are reliable and the research is well organized. However, several comments should be addressed:
1. I suggest the authors point the specific time of the land use area in line 95-96, due to the proportion was different in different years.
2. How to determine the weight of vulnerability indicators? Is it reasonable?
3. Some reasons or references are needed to clarify the vulnerability selection indicators.
4. It is suggested that the author should reduce the repetition of the same sentences (line140-145), and use summary sentences to explain the phenomena and research results as much as possible. The expression of some keywords is inconsistent, such as “landscape ecological risk index”, “landscape ecological risk”.
5. Is it reasonable to classify the ecological risk level into five levels when calculating the temporal and spatial variation of LERI?
6. The article should pay attention to the use of academic terms, such as natural interruption classification method should be Natural Breaks. It is suggested that the authors comb the whole paper and correct it.
7. The figures should be stand alone, so I suggest the authors reorganize the figure name which should be include the study area and period.
8. In the references list, the name of some periodicals used their abbreviations, while some used the complete spelling. It should be unified according to the demand of IJERPH.
9. Minor errors: problem of capital and lower-case letter in line 61; the style of north arrow in different figures should be unified; no space before paragraph in line 151 and 157; there should be one space between digit and unit, e.g. line 166. These minor errors should be checked through the paper.
Author Response
Thank you very much for your comments, details of our point-to-point revision are attached

Reviewer 2 Report
I am very interested in taking the Yellow River Ecological Economic Belt as the study area, enhancing the ecological risk assessment framework and indicators from the perspective of landscape, assessing the temporal and spatial differences of ecological risks, and determining the key areas for managing ecological risks. There is much to be learned about this topic. I have, however, the following concerns:
1. The major issues.
(1) There is a great deal of clarity in the author's abstract. However, I believe that it is possible to refine it further?
(2) Could the author add some content at the end of the introduction? The article structure is added here.
(3) In the section of 2.1. Study area, would you be able to add the data in this section to the source references in order to ensure the authenticity and credibility of the data in the paper?
(4) The results should be logically-structured, involving the main and interesting conclusions.
2. The minor issues.
(1) The tenses of the paper should be carefully checked. In particular, the line of 33, 82, and 122.
(2) Please check carefully whether the initial capitals are used correctly in the paper. Especially, the line of 33 and 82.
(3) When editing a formula, it is recommended to use Math-type to ensure a more standardized expression.
(4) In Table 2, please check carefully whether some statements contain spaces.
(5) Please check whether the line spacing is used correctly in the paper. Especially, in Table 2 and the line of 139, 77 and 247.
(6) In Figure 5 and Figure 8, I find it lacking in clarity. The author may be able to make the picture clearer in the paper in order to ensure a high standard.
(7) I would appreciate it if you checked the details carefully. Please pay particular attention to lines 16, 157, 186, 318 and 385.
(8) Please ensure that each reference is formatted correctly. There were some references that lacked page numbers and DOI. In addition, the journal name of the cited reference should also be abbreviated. Furthermore, please make sure that some references correctly format the author's name. In the end, a Chinese journal is marked if it is cited.
(9) A review by a native English speaker is required in order to make the paper more intelligible and easier to follow.
Author Response

(The authors gave the same response as above.)

Reviewer 3 Report
This study improved the framework and indicators for ecological risk evaluation from the perspective of landscape, evaluated the spatial and temporal differences of ecological risk, and identified the key areas for ecological risk management, on the example of the Yellow River Ecological Economic Belt as study area. The manuscript is clear, relevant for the field and presented in a well-structured manner, using the figures/tables/images/schemes appropriately. The conclusions are consistent with the results and methodological approaches presented. I have just few specific comments:
-
Study area - provide more information about natural condition of study area, e.g. specify diverse landform types, result descriptions mentions Qinghai Lake, Qilian Mountain - provide more details about landscape character
-
Fig.2 - I recommend completing the picture, highlighting the relationships between data that enter into analyses and evaluations, add abbreviations of indices described in the detailed description of the methodology
-
ambiguities in the formula (line 130 and 133) - landscape disturbance index ??? or difference between ????
-
Landscape vulnera-bility index (Table 2) - "values to different land use types through the expert scoring method"- it would be appropriate to confront the assigned values with other scientific studies, I doubt above value cropland = 4, grassland = 3,wood-land = 2,and urban land = 1
-
Weighted sum of indicators(???)(Table 2) - discrepancies in formula and description ???
-
Fig 3a - I recommend to change chart type, as changes in units are not very visible
-
line 198-201 - description of methodological approach
Author Response

(The authors gave the same response as above.)

Round 2
Reviewer 2 Report
1. In the section on Conclusion, would it be possible to include a comparison between the results of this study and those of previous similar studies?
2. The conclution should prcede the limitation.
3. Suggest to provide the clean version
Author Response
Thank you very much for your comments, details of our point-to-point revision are attached.
